# A Proposal for a User-Oriented Spatial Metadata Profile

**Mohsen Kalantari [1,\*], Syahrudin Syahrudin [2], Abbas Rajabifard [1] and Hannah Hubbard [1]**

1   Department of Infrastructure Engineering, The University of Melbourne, Melbourne, VIC 3010, Australia; abbas.r@unimelb.edu.au (A.R.); hubbardh@unimelb.edu.au (H.H.)
2   Indonesian Geospatial Agency, Jakarta 16911, Indonesia; syahrudin@big.go.id
\*   Correspondence: mohsen.kalantari@unimelb.edu.au

**Abstract:** Spatial metadata profiles have been designed and evolved by data custodians to manage, share, discover, and use spatial data. The end-users of spatial data often do not have much input in designing the profiles. The spatial data infrastructure literature reveals that they question the usability of spatial metadata. This paper analyzes the usability of metadata profiles by engaging end-users and clarifying their requirements in response to this problem. Over 60 users from 18 countries were engaged using an online survey based on a purposive sampling method. The results show that the most widely used metadata standard, ISO 19115, provides metadata elements to accommodate most user requirements for searches. However, an extension to the standard is necessary to assist users in discovery and selection. Two new metadata elements are proposed as part of the extension. The extension also involves changing the obligation type of existing elements to improve data discovery.

**Keywords:** spatial data infrastructure; spatial metadata; user-centered design; ISO 19115; spatial data portals

## 1. Introduction

Spatial data custodians have predominantly designed metadata systems and standards to manage and make sharing, discovery, and usage of data possible. Most spatial metadata records contain descriptions about spatial data following a metadata standard such as ISO 19115. The latest version of ISO 19115-1:2014 is available as ISO 19115-1:2014/Amd 2:2020. ISO standards undergo a review every five years. The standard provides a framework that specifies how metadata should be created (e.g., by dictating metadata elements and classes of information). The metadata are published for spatial data users via web catalogue systems or spatial data portals that help users find, select, and acquire spatial data for their projects or applications.

However, the usability of spatial metadata records is questioned. The standards were initially designed for spatial data producers for inventory purposes, and not for use [1–3]. Moreover, with the growing number of spatial data users outside the spatial field, the needs and requirements of non-experts create new expectations for metadata records and spatial data portals.

A substantial body of research has addressed the problems of creating complete and consistent metadata records [4–12]. The outcomes of these research works offer automatic approaches to create and update metadata records. There is also existing research on user involvement in metadata creation and improving spatial data discovery using semantics and ontologies [13–22]. These papers take state-of-the-art to the next level and argue that the role of users in the creation, organization, and even discovery of metadata records should be augmented.

This article complements the existing body of research and addresses usability problems. The paper identifies user needs and expectations from spatial metadata for discovery and selection. In Section 2, it engages with end-users to determine their requirements. In Section 3, it maps the needs and expectations of users to metadata elements in ISO 19115

and determines if the standard complies with the user requirements. A user-oriented spatial metadata profile, as an extension to the standard, is developed in Section 4. The limitations of the research are presented in Section 5, and a conclusion is reached in Section 6.

## 2. Research Method

A previous study on the usability of spatial data portals identified gaps regarding the effectiveness and efficiency of metadata systems in Australia [23]. The study revealed inconsistencies and issues in the metadata records and also identified improvement areas for user interfaces.

Building on the outcomes of [23], a questionnaire was developed to engage an international audience and gauge their opinions about the spatial data portals they utilize. Purposive sampling was implemented for the sample selection by conducting snowball sampling worldwide. Spatial data users known to the authors were contacted and asked to be participants. The authors circulated the questionnaire internationally via email to their known associates. These people were asked to pass the questionnaire to their own associates who shared similar qualifications. The authors circulated the questionnaire via LinkedIn and Twitter. Selected social media accounts belonging to spatial communities and organizations were also tagged. To ensure participant validity with this open distribution method, several questions were designed to obtain their backgrounds and experiences and detect whether or not they were suitable for this survey. Following this research's human ethics requirements, we did not ask for the identities of the portals to avoid potential exposure to the study participants.

As illustrated in Table 1, the questionnaire contained five sections: Introduction, Participant's Information, Spatial Data Search and Discovery, Title and Abstract as well as Spatial Data Suitability Assessment and Selection. The Title and Abstract section was specifically designed to gather information from the survey participants relating to their experiences and opinions concerning the content, number of words, and the presentation of titles and abstracts shown in the search results. Titles and abstracts are key elements related to the usability of metadata for discovery and selection [23]. Other survey questions were developed based on [23] and ISO 19115. In Table 1, searching criteria refer to a broad range of items including content/type of dataset, geographic coverage, production and update dates, scale, format, and producer. We also gave an option to the participants to provide their set of criteria.

**Table 1.** Structure of the research questionnaire regarding the user needs and expectations for spatial metadata.

| Section | Topics | Sub-Topics |
|---------|--------|------------|
| 1 | Introduction | ✓ Introduction to research and questionnaire<br>✓ Plain language statement<br>✓ Consent form |
| 2 | Participant's information | ✓ Working field<br>✓ Level of spatial knowledge<br>✓ Years of experience<br>✓ Location (country)<br>✓ Spatial data familiarity<br>✓ Spatial metadata familiarity |
| 3 | Spatial data search and discovery | ✓ Searching criteria<br>✓ Interfaces to interact for searching |

**Table 1.** *Cont.*

| Section | Topics | Sub-Topics | |
|---|---|---|---|
| 4 | Title and abstract | ✓ | Content |
| | | ✓ | Number of words |
| | | ✓ | Interfaces to present Title and Abstract |
| 5 | Spatial data suitability assessment and selection | ✓ | Content |
| | | ✓ | Number of words |
| | | ✓ | Interfaces to assess and select data |

The questions were tested and refined internally and externally. Internal testing was implemented by conducting a series of evaluations testing the questions against the objectives and ensuring worldwide accessibility using any Internet browser and any device. The questionnaire was also tested to ensure that responses were appropriately recorded. External testing was implemented by sending the questionnaire to the researchers' associates for feedback. The feedback was incorporated to refine the questions.

## 3. Results

The survey was open for two months. Sixty-one participants from 18 different countries (Australia, Austria, Azerbaijan, Canada, Colombia, Germany, Indonesia, Iran, Italy, Jamaica, Mozambique, New Zealand, Pakistan, Singapore, South Africa, Thailand, the Netherlands, and the USA) responded to the questionnaire. The following sections describe the results of the survey, starting with the participants' demographics.

### 3.1. Participants' Demographics

Information regarding the participants' demographics was extracted from the responses to Section 1 of the questionnaire (Participant's Information). The first set of data pulled from Section 1 is related to the field of work or sectors where the participants spend most of their professional time, as illustrated in Table 2.

**Table 2.** Participants' fields of expertise.

| No. | Field of Work | Number of Participants |
|---|---|---|
| 1 | Public health | 2 |
| 2 | Auditor | 1 |
| 3 | Biodiversity | 1 |
| 4 | Civil engineering | 2 |
| 5 | Consumer goods | 1 |
| 6 | Data analytics | 1 |
| 7 | Disaster management | 1 |
| 8 | Environment | 2 |
| 9 | Financial | 1 |
| 10 | Geodesy | 1 |
| 11 | Geospatial data production | 1 |
| 12 | GIS or spatial services | 32 |
| 13 | GNSS positioning | 1 |
| 14 | Higher education | 1 |
| 15 | Land management and cadaster | 6 |
| 16 | Planning | 4 |
| 17 | Psychology | 1 |
| 18 | Remote sensing | 1 |
| 19 | Social science | 1 |

These results show that the participants came from diverse professional backgrounds. However, most of the participants worked in GIS or spatial services, where spatial data and information are utilized regularly. Users who apply their spatial knowledge in the other

domains listed are regarded as experts. The results indicate that there are knowledgeable and novice users outside the spatial industry in the survey.

In addition to gathering data about the participants' professional backgrounds, the survey also sought to discover their levels of spatial knowledge and familiarity with spatial data, as illustrated in Table 3.

**Table 3.** Participants' levels of spatial knowledge.

| Participants' Responses | Category | Number |
|---|---|---|
| I sometimes work with spatial data and have a general understanding of spatial knowledge | Knowledgeable | 1 |
| I work with spatial data and have a general understanding of spatial knowledge | Knowledgeable | 17 |
| I have never worked with spatial data or information before | Novice | 2 |
| I work with spatial data and have an extensive understanding of spatial knowledge at a high level | Experts | 37 |
| I work with spatial data, and I have academic or professional training in spatial science | Experts | 1 |
| I work with spatial data, but I do not have academic or professional training in spatial science | Knowledgeable | 3 |

From Table 3, it can be seen that 38 of the participants regarded themselves as experts, 21 identified themselves as knowledgeable and sometimes working with spatial data, and two participants answered they have had no spatial training or education and have never worked with spatial data.

The participants' levels of experience in using spatial data portals or websites is also helpful information in analyzing their spatial data discovery and selection responses. This information can be seen in Table 4.

**Table 4.** Participants' experience with spatial data portals or websites.

| Participants' Responses | Category | Number |
|---|---|---|
| I have never searched for spatial data from web spatial data catalogues or portals. | None | 5 |
| I have searched for spatial data from web catalogues for some projects in the past. | Occasional | 30 |
| I regularly search for spatial data from web data catalogues or data portals. | Regular | 26 |

From Table 4, it can be seen that 26 of the participants were regular users of spatial data portals (making their responses particularly valuable for developing user-oriented metadata profiles), 30 occasionally used the portals, and five have never used them. The responses enabled us to explore what users perceived as their needs.

### 3.2. Spatial Data Discovery

A closed-ended question about search criteria such as content, coverage, scale, and format was presented. Open-ended questions were also given to participants in order to understand their preferred interfaces for searches such as dropdown lists and freehand drawing tools and allowed them to explain their answers. Their responses were categorized based on their levels of spatial knowledge, as shown in Tables 5–7.

**Table 5.** Novice participants' preferred interface for spatial data discovery.

| No. | Searching Criteria | Responses | User Interface |
|---|---|---|---|
| 1 | Data content | 2 | Dropdown list |
| 2 | Geographic coverage | 2 | Dropdown list |
| 3 | Time-related content | 1 | Dropdown list |

**Table 6.** Knowledgeable participants' preferred interface for spatial data discovery.

| No. | Searching Criteria | Responses | User Interface |
|---|---|---|---|
| 1 | Data content | 21 | Free text, predefined text, dropdown list |
| 2 | Geographic coverage | 17 | Freehand drawing tool, coordinates, geographic names, buffer tool |
| 3 | Time-related content | 12 | Dropdown list, free text |
| 4 | Data format | 10 | Dropdown list, checklist |
| 5 | Spatial scale/resolution | 9 | Dropdown list, checklist, free text |
| 6 | Data producer | 6 | Dropdown list, checklist |

**Table 7.** Expert participants' preferred interface for spatial data discovery.

| No. | Searching Criteria | Responses | User Interface |
|---|---|---|---|
| 1 | Data content | 33 | Free text, predefined text (catalogue tree), dropdown list, map preview |
| 2 | Geographic coverage | 35 | Freehand drawing tool, coordinates, geographic names, buffer tool, map preview |
| 3 | Time-related content | 20 | Dropdown list, free text |
| 4 | Data format | 22 | Dropdown list, checklist |
| 5 | Spatial scale/resolution | 19 | Dropdown list, checklist, free text |
| 6 | Data producer | 12 | Dropdown list, checklist |

The above tables show that both knowledgeable and expert users require detailed information to discover potential spatial data. The access to more specific criteria allows knowledgeable and expert users to narrow down their searches, tailoring them more efficiently to their particular needs. Based on these participants' responses, criteria for ***knowledgeable and expert users'*** searches need to include:

- What the data is about;
- The geographical area that the data covers;
- When the data was created and/or updated;
- The quality of the data; and
- Who the data has been produced by.

As the above tables show, the novice participants gave different answers from the knowledgeable and expert participants, only requiring search criteria to cover information relating to the data, its coverage, and the date it was created and/or updated. The novices' responses also omitted the need for search criteria that would enable them to search for information concerning the quality of the data or the data producer.

The questionnaire also included questions designed to gather information about the users' preferred interface for each criterion. As shown in Table 8, the participants preferred dropdown lists, check boxes, or free text for text-based criteria. There were several alternatives favored by the participants for geographic coverage such as freehand drawing, coordinates, and geographic names.

The results also indicate that more detailed interfaces were preferred by the participants. These preferences can be grouped based on how frequently participants search for data. As shown in Table 8, the participants who used spatial data portals occasionally and those who used them regularly shared similar preferences regarding how the search criteria were presented and selected. Dropdown lists appeared to be the most preferred interface. Most participants from the occasional and regular user groups preferred a freehand boundary for location-based search criteria.

**Table 8.** User interface preferences for the search criteria.

| Frequency/Regularity of Use of Spatial Data Portals | No Experience | Occasional User | Regular User |
|---|---|---|---|
| The preferred interface for search criteria | ✓ Dropdown list | ✓ Geographic names<br>✓ Free text for content<br>✓ The bounding box for location searches<br>✓ Dropdown list<br>✓ A freehand boundary for location searches<br>✓ Dropdown list<br>✓ Filter-based presentation as in e-commerce<br>✓ Free text for location searches<br>✓ Buffer using a coordinate for location searches | ✓ The bounding box for location searches<br>✓ Dropdown list<br>✓ A freehand boundary for location searches<br>✓ Check box<br>✓ Multiple choice<br>✓ Catalogue tree for content |

### 3.3. Title and Abstract

As mentioned in an earlier section, the questionnaire responses indicate that titles and abstracts displayed in the search results are fundamental for spatial data discovery and selection. Spatial data users might decide whether they continue or stop the search process based on the title and abstract information. Inconsistency and irrelevant presentation of information in titles and abstracts hamper discovery and selection [23]. The questionnaire, therefore, consists of questions specifically designed to address these problems. Participants' responses to these questions are shown in the following tables.

As illustrated in Table 9, the participants' responses indicated what data were preferred in titles on the results pages. They voted that titles should contain the preferred data in no more than ten words, as illustrated in Table 10.

**Table 9.** Participants' responses concerning content for the titles.

| Participant Category | Novice | Knowledgeable | Expert |
|---|---|---|---|
| Content desired in titles | ✓ Data content (2)<br>✓ Geographic coverage (2) | ✓ Data content (22)<br>✓ Geographic coverage (14)<br>✓ Production date (10)<br>✓ Spatial scale/resolution (10)<br>✓ Data format (9)<br>✓ Last update (4) | ✓ Data content (37)<br>✓ Geographic coverage (32)<br>✓ Production date (15)<br>✓ Last update (13)<br>✓ Data format (23)<br>✓ Spatial scale/resolution (21) |

**Table 10.** Participants' responses concerning the number of words for the titles.

| Up to 5 | Up to 10 | Up to 15 | Up to 20 | No Limit |
|---|---|---|---|---|
| 10 responses | 27 responses | 10 responses | 6 responses | 8 responses |

The participants also gave responses indicating their preferences regarding the information in the abstracts. Knowledgeable and expert users gave very similar responses in this regard, as seen in Table 11. The noticeable difference between the two groups was their preferences regarding the production process: only two participants from the knowledgeable user group indicated a preference for this information compared to 14 participants from the expert group. The results suggest that expert users preferred more detailed information related to the spatial data to verify the data quality against the quality declared in the relevant element.

**Table 11.** Participants' responses concerning information content for the abstracts.

| Novice | Knowledgeable | Expert |
|---|---|---|
| ✓ Content description—explanation about the feature and its attributes (1)<br>✓ Detailed explanation about geographic (location) coverage/extent (1)<br>✓ Spatial scale/resolution (1)<br>✓ Production date (2)<br>✓ Data accuracy or error (1)<br>✓ Level of detail or generalization of the feature presented in the data, e.g. the lowest road type (1)<br>✓ Intended use of the data (1)<br>✓ Production process (1) | ✓ Content description—explanation about the feature and its attributes (20)<br>✓ Spatial scale/resolution (14)<br>✓ Production date (12)<br>✓ Detailed explanation about geographic (location) coverage/extent (11)<br>✓ Data format (10)<br>✓ Data accuracy or error (10)<br>✓ Maintenance date (9)<br>✓ Intended use of the data (8)<br>✓ Level of detail or generalization of the feature presented in the data, e.g. the lowest road type (6)<br>✓ Production process (2) | ✓ Content description—explanation about the feature and its attributes (34)<br>✓ Detailed explanation about geographic (location) coverage/extent (29)<br>✓ Data format (25)<br>✓ Spatial scale/resolution (23)<br>✓ Data accuracy or error (22)<br>✓ Production date (19)<br>✓ Level of detail or generalization of the feature presented in the data, e.g. the lowest road type (19)<br>✓ Intended use of the data (15)<br>✓ Maintenance date (14)<br>✓ Production process (14) |

To maintain the consistency and readability of the abstracts, participants were asked to vote on the maximum number of words in order for the authors to identify the characteristics of a consistent and readable abstract. As shown in Table 12, most participants voted for a maximum of 150 words in an abstract. Considering the preferred content of abstracts as selected by the participants, it is reasonable to expect an abstract of this word count to present all the required information.

**Table 12.** Participants' responses concerning the number of words for the abstracts.

| Up to 100 | Up to 150 | Up to 250 | Up to 400 | No Limit |
|---|---|---|---|---|
| 8 responses | 18 responses | 13 responses | 7 responses | 16 responses |

In addition to the information provided in response to the closed-ended questions, survey participants were also invited to respond to open-ended questions regarding the search results pages. These questions asked participants to indicate any preferences for additional information and provide any further explanations for their answers. The participants' responses to these open-ended questions are shown in Table 13.

**Table 13.** Additional information concerning the search results pages.

| No. | Item | Description |
|---|---|---|
| 1 | User ratings, data citation | As an online product, users' ratings for each data set might be helpful to get an insight into the quality or usability of the data |
| 2 | Thumbnail, pop-up image, graphical presentation by Quicklook | A graphical Quicklook of the spatial data showing the extent of the data |
| 3 | Keywords | Keywords related to the data |
| 4 | License | The intellectual property of the data that limits its use |
| 5 | Organization | Data producer or owner |
| 6 | Number of downloads | Number of downloads by previous users |
| 7 | Location | The geographic location of the data relative to the surrounding area |
| 8 | Examples | A small portion of the data that can be downloaded or accessed |
| 9 | Data preview, graphics preview | Map viewer for spatial data or chart/graph viewer of non-spatial data |

It should be noted that only the experts gave their opinions on the open-ended questions. Their responses indicate that they prefer to find out details related to the datasets as early as possible to identify the potential and relevant data for their applications.

They suggest that sample data and previews should accompany the titles and abstracts so that the suitability of the data for their applications can be determined as early as possible.

### 3.4. Spatial Data Selection

The responses to the spatial data selection questions showed the same pattern as the responses to spatial data discovery, as shown in Table 14. The results indicate that novice users are only interested in what the data are about as they do not possess sufficient knowledge to assess the data's suitability. Knowledgeable and expert users, on the other hand, prefer detailed information about the spatial data. Information about the content, geographic coverage, attributes, accuracy, or error remains a significant consideration for these users. Only a few participants considered previous uses (by other users) for the selection. This might indicate that the knowledgeable and expert users select spatial data based on their assessment of the data, rather than the experiences of others.

**Table 14.** Information required by participants for spatial data selection.

| Novice | | Knowledgeable | | | Expert | | |
|---|---|---|---|---|---|---|---|
| ✓ | Feature/theme | ✓ | Feature/theme | 16 | ✓ | Feature/theme | 16 |
| | | ✓ | Geographic coverage/extent | 16 | ✓ | Feature attribute description | 16 |
| | | ✓ | Last update (maintenance) | 14 | ✓ | Geographic coverage/extent | 14 |
| | | ✓ | Data format | 14 | ✓ | Resolution/pixel size | 14 |
| | | ✓ | Production date | 13 | ✓ | Data format | 13 |
| | | ✓ | Spatial scale | 13 | ✓ | Spatial scale | 13 |
| | | ✓ | Resolution/pixel size | 12 | ✓ | Data type | 12 |
| | | ✓ | Data producer | 12 | ✓ | Use restriction | 12 |
| | | ✓ | Use restriction | 12 | ✓ | Positional accuracy/error | 12 |
| | | ✓ | Data type | 11 | ✓ | Price | 11 |
| | | ✓ | Price | 11 | ✓ | Production date | 11 |
| | | ✓ | Owner contacts | 11 | ✓ | Last update (maintenance) | 11 |
| | | ✓ | Feature attribute description | 10 | ✓ | Attribute accuracy/error | 10 |
| | | ✓ | Positional accuracy/error | 9 | ✓ | Owner contacts | 9 |
| | | ✓ | Topology | 8 | ✓ | Level of detail/generalization | 8 |
| | | ✓ | Production process/history | 7 | ✓ | Data producer | 7 |
| | | ✓ | Intended use | 7 | ✓ | Topology | 7 |
| | | ✓ | Level of detail/generalization | 6 | ✓ | Production process/history | 6 |
| | | ✓ | Attribute accuracy/error | 5 | ✓ | Intended use | 5 |
| | | ✓ | Data sample | 5 | ✓ | Data sample | 5 |
| | | ✓ | Previous uses | 3 | ✓ | Previous uses | 3 |

The knowledgeable and expert participants also required additional information for spatial data selection, mainly related to their preference for assessing the data themselves. This additional information includes, for example, access to relevant data or samples and map services for preview. Table 15 shows the other requirements indicated by the participants.

**Table 15.** Additional information required for spatial data selection.

| Additional Information for Data Selection |
| --- |
| ✓     Link to relevant data (e.g., link to ID of mosaicked orthoimages) |
| ✓     Graphical preview of the data |
| ✓     Data sample containing complete attributes and domains |
| ✓     Graphical representation of the spatial extent or footprint |
| ✓     Licensing (e.g., CC) for commercial use, etc., perhaps covered by use restriction |
| ✓     A graphical view of the data in a web map |
| ✓     Technical constraints |
| ✓     Update frequency |
| ✓     Variable/attribute information |
| ✓     Copyright or compliance use of the data |
| ✓     Data assurance in a graphical form |
| ✓     Future data updates (as planned) |
| ✓     Utilization and usefulness of the data |

The additional requirements for spatial data selection were similar to the additional requirements for data discovery. This suggests that the participants preferred to get the information as early as possible and assess the data by looking at the sample or preview after finding the potential data from the title and abstract.

The results also identified the participants' user interface preferences: e-commerce or an online shopping style for the portal, and a mapping service for data preview. One participant from the regular user group specifically mentioned that they were upset with the functionality of the portal they used for searching and selecting spatial data. The participant indicated that it would be beneficial if the portal had a window showing relevant data or previous search information so that they would not have to do another search or go back to the results pages every time. The above-mentioned information required by participants for data discovery and selection including the preferred interface for data selection is the basis for developing a user-oriented spatial metadata profile, as explained in the following sections.

## 4. User-Oriented Spatial Metadata Profile Development

Having a list of requirements for spatial data users including the preferred user interfaces for discovery and selection provides the information needed to redesign and redevelop spatial metadata and user interfaces. Spatial metadata systems are created following a standard such as ISO 19115. There is a requirement for the systems to be developed according to the standard. Similarly, it is essential to investigate how, or to what extent, the standard meets the user requirements and expectations identified by the questionnaire results. Mapping the user requirements against the standard can start this process.

### 4.1. Mapping the Metadata Standard with the User Requirements and Expectations

The latest version of the ISO standard describing fundamental geospatial metadata elements is ISO 19115-1:2014/Amd 2:2020, Geographic information—Metadata, Part 1: Fundamentals. According to the standard, full metadata is an aggregate of 12 metadata classes. Within the classes, there are metadata elements that contain information related to spatial data characteristics. This information can be used to answer the questions of 'what', 'when', 'where', 'who', 'why', and 'how' in relation to spatial data or resources. Table 16 shows the user requirements and the relevant metadata elements from ISO 19115-1:2014.

**Table 16.** Mapping user requirements to ISO 19115-1:2014 elements.

| User Requirements | ISO 19115-1:2014 Elements | Obligation/ Maximum Occurrence/ * |
|---|---|---|
| Title | MD_DataIdentification>CI_Citation.title | M/1 |
| Abstract | MD_DataIdentification.abstract | M/1 |
| Content/theme | MD_DataIdentification>MD_TopicCategoryCode | C/N |
| Geographic coverage | MD_DataIdentification>EX_Extent> EX_GeographicExtent. geographicBoundingBox | C/N |
| Geographic name | MD_DataIdentification>EX_Extent> EX_GeographicExtent.geographicDescription | C/N |
| Data format (spatial rep) | MD_SpatialRepresentation> MD_SpatialRepresentationTypeCode | O/N |
| Data type | MD_Metadata>MD_MetadataScope. resourceScope>MD_ScopeCode | C/1 |
| Scale | MD_DataIdentification.spatialResolution> MD_Resolution.equivalentScale | O/N |
| Resolution | MD_DataIdentification.spatialResolution> MD_Resolution.distance | O/N |
| Positional accuracy | <<ISO19157>>DQ_Result>DQ_QuantitativeResult | M/N |
|  | <<ISO19157>>DQ_Result>DQ_DescriptiveResult.statement | M/1 |
| Attribute accuracy | <<ISO19157>>DQ_Result>DQ_QuantitativeResult | M/N |
|  | <<ISO19157>>DQ_Result>DQ_DescriptiveResult.statement | M/1 |
| Level of detail | MD_Resolution.levelOfDetail | C/1 |
| Data assurance | <<ISO19157>>DQ_Result>DQ_ConformanceResult.explanation | O/1 |
| Production date | MD_DataIdentification>CI_Citation>CI_Date> DateTypeCode | O/N |
| Last update | MD_DataIdentification>MD_MaintenanceInformation. maintenanceDate>DateTypeCode | O/N |
| Maintenance schedule | MD_DataIdentification>MD_MaintenanceInformation. maintenanceAndUpdateFrequency>MD_MaintenanceFrequencyCode | C/1 |
| Producer | MD_DataIdentification>CI_Citation>CI_Responsibility. role>CI_RoleCode | O/N |
| Intended use | MD_DataIdentification.purpose | O/1 |
| User ratings | No corresponding or relevant ISO element |  |
| Number of downloads (use) | No corresponding or relevant ISO element |  |
| Previous uses | MD_DataIdentification>MD_Usage.specificUsage | M/1 |

**Table 16.** *Cont.*

| User Requirements | ISO 19115-1:2014 Elements | Obligation/ Maximum Occurrence/ * |
|---|---|---|
| User reviews | MD_DataIdentification>MD_Usage.identifiedIssues | O/N/ * |
| | MD_DataIdentification>MD_Usage.specificUsage | M/1 |
| Utilization/usefulness | MD_DataIdentification>MD_Usage.identifiedIssues | O/N/ * |
| | MD_DataIdentification>MD_Usage.specificUsage | M/1/ * |
| | <<ISO19157>>DQ_Result>DQ_ConformanceResult.explanation | O/1/ * |
| Thumbnail | MD_DataIdentification>MD_BrowseGraphic.fileName | M/1 |
| | MD_DataIdentification>MD_BrowseGraphic.linkage | O/N |
| Keywords | MD_DataIdentification>CI_Citation.otherCitationDetails | O/N |
| | MD_DataIdentification.descriptiveKeywords>MD_Keywords | O/N |
| Data sample | MD_DataIdentification>CI_Citation. onlineResource | O/N |
| Data viewer | MD_DataIdentification>CI_Citation. onlineResource | O/N |
| Attribute | MD_CoverageDescription>MD_AttributeGroup.contentType | M/N |
| | MD_Metadata>MD_ContentInformation>MD_CoverageDescription | M/1 |
| License (legal restrictions) | MD_DataIdentification>MD_Constraints.graphic | O/N |
| | MD_DataIdentification>MD_LegalConstraints | O/N |
| | MD_DataIdentification>MD_SecurityConstraints | O/N |
| Use limitation (technical) | MD_DataIdentification>MD_Constraints.useLimitation | O/N |
| | MD_DataIdentification>MD_Usage.userDeterminedLimitations | O/N |
| Price | MD_Distributor>MD_StandardOrderProcess.fees | O/1 |
| Production process | MD_Metadata>LI_Lineage | O/N |
| Topology | MD_SpatialRepresentation>MD_VectorSpatialRepresentation.topologyLevel>MD_TopologyLevelCode | O/1 |
| Owner/distributor contact | MD_Distribution>MD_Distributor.distributorContact | M/1 |
| | MD_DataIdentification>CI_Citation.citedResponsibleParty | O/N |

* Potentially relevant to the corresponding ISO element. M: Mandatory. O: Optional. C: Conditional. N: Infinite occurrence.

As shown in Table 16, there are two types of information in the list of user requirements that the standard does not currently have related elements for: user ratings and the number of downloads. Two other requirements—user reviews and data usefulness—do not have identical corresponding elements, but there are established elements that are potentially suitable for them. However, the standard has been prepared to accommodate data use/utilization information by providing MD_Usage subclasses under the MD_Identification class. The profile can propose new elements for ratings and the number of downloads under the subclass as follows:

MD_DataIdentification>MD_Usage.numberOfUsage—M/1
MD_DataIdentification>MD_Usage.userDefinedRating>MD_UserRatingCode—M/1

Other requirements that appeared in the survey results were data sample and data preview. These requirements reflect the capacity of participants to assess the suitability of data by directly exploring and reviewing the data instead of just relying on the metadata. Both requirements can be accommodated by providing corresponding links to download or access the sample or browse the data via a web mapping service in the OnlineResource element under the CI_Citation subclass.

Apart from the unavailable elements for the above-mentioned user-related information, most of the requirements are accommodated by the standard. Interestingly, the usability evaluation results obtained by [23] revealed missing information, inconsistency, and irrelevance problems in keywords, titles, and abstracts.

Missing information can result from different types of obligation for the element in the standard or of non-existent information. When optional, an obligation being assigned to an element means that the author may opt to provide the information or to not do so. Setting a mandatory type for user-required elements can encourage authors or responsible parties for specific data to make the information available in the first place. Information being non-existent might occur due to the implementation of minimum/core metadata in the standard. Minimum/core metadata are defined as a set of elements that can be used for both metadata management and data discovery. It is designed to help the data producers create metadata for their data inventory. However, the information/element in the minimum metadata may not necessarily suffice for discovering and selecting the data.

The irrelevance and inconsistency problems identified in the evaluation are mainly found in titles and abstracts. The potential source of the problems is that both titles and abstracts are assigned with a free text domain. Although the standard defines titles as 'the name by which the cited resource is known', this is still too vague to ensure that the author will provide the relevant information. This also applies to abstracts, defined by the standard as 'a brief narrative summary of the resource'. The problems can be addressed by assigning strict rules or guidelines in the standard for authors or machines to abide by when creating titles, abstracts, and other elements when possible. Such rules or guidelines can dictate what information should be presented in the titles and abstracts as well as in what order. They can also dictate the maximum number of words allowed in both the titles and abstracts.

The Spatial Information Council of Australia and New Zealand (ANZLIC) developed their metadata profile with guidelines for organizations to create their metadata including detailed explanations for every element. However, the guidelines are vague and do not prevent inconsistency in the metadata records. They do not provide clear instructions to ensure consistency and clarity in the content (e.g., in titles and abstracts), as can be seen in Tables 17 and 18.

**Table 17.** Abstract description and guidance in the ANZLIC Metadata Profile Guidelines Version 1.2 [24].

| Name | Abstract |
|------|----------|
| Definition | Brief narrative summary of the content of the resource |
| Data type | CharacterString |
| Domain | Free text |
| Meaning & Purpose | The identification abstract provides additional information about the resource. This may allow users to obtain a better appreciation of the resource and assist them to determine fitness for purpose |
| Guidance | The abstract should provide sufficient information such as keywords to adequately describe the content of the resource. Careful consideration should be given when preparing an abstract as it is an important element for the assessment of a resource |

**Table 18.** Title description and guidance in the ANZLIC Metadata Profile Guidelines Version 1.2 [24].

| Name | Title |
|------|-------|
| Definition | Name by which the cited resource is known |
| Data type | CharacterString |
| Domain | Free text |
| Meaning & Purpose | The resource title is the official name for the resource. Where no normal name for the resource, a useful name for the resource should be assigned. If the resource is a text document, use the full title as it appears on the title page; otherwise use a meaningful, plain language phrase for that resource (i.e., do not use the file name) |
| Guidance | The title naming conventions should be consistently used for related resources (e.g., to facilitate discovery). To discriminate between duplicate titles, a reference to the version should be included in the title. For identification purposes, it is important to carefully complete this element. Other users should easily understand the title. If the resource is known by an alternate title, include this in the alternate title element. |

The rules for ensuring clarity and consistency in titles and metadata can be inducted in the metadata standards by assigning a special domain for them or keeping the free text domain and including the required information in the definition. This domain would aggregate the content of selected metadata elements including their order, as can be seen in Table 19.

**Table 19.** Proposed information in the domain/definition for the titles and abstracts in a user-oriented profile.

| Element | Data Type | Domain or Definition |
|---------|-----------|----------------------|
| Title | CharacterString | <<Content>><<GeographicName>><<Scale/Resolution>> <<Production.Date>><<LastUpdate>> <<SpatialRepresentation>> |
| Abstract | CharacterString | <<Content>><<GeographicName>><<AttributeDescription>> <<CoverageDescription>><<LevelOfDetail>> <<PositionalandThematicAccuracy>><<Lineage>> <<IntendedUse>><<Maintenance>> |

The element mapping results explained above reveal that ISO 19115 accommodates the most user-required information, with the exception of user-defined information (e.g., user data ratings and user reviews). New elements should be introduced to the profile to allow for the information to be entered in the metadata records. The results also revealed that the standard, while able to ensure the consistency of the metadata record structure, failed to guarantee consistency and clarity in the information provided in key elements of

the metadata (e.g., in titles and abstracts) due to the effect of the free text domain with a general definition.

The user-oriented spatial metadata profile proposed by this paper is expected to fill the gaps in ISO 19115 by extending the standard while following the extension guidance provided in it, as explained in the next section.

### 4.2. Extension of ISO 19115:2014 Discovery Metadata for Geospatial Resources

The user-oriented spatial metadata profile is expected to provide rules and guidelines to redesign and redevelop metadata records and user interfaces. This redesign and redevelopment will be aimed at improving the usability of metadata for spatial data discovery and selection.

The profile is essentially an extension of the ISO 19115 metadata standard to incorporate the users' information to help them discover and select spatial metadata. In this paper, the term 'profile' does not refer to the community profile described in the ISO 19115-1:2014 document. This paper specifically extends the discovery metadata for geographic resources, as in Annex F in the standard document. Therefore, it follows the rules for creating an extension as given in Annex C in the standard. According to these rules, an extension is allowed if the standard does not accommodate a specific requirement, and one or more of the following extension types is allowed:

1. adding a new metadata package;
2. creating a new metadata codelist to replace the free text domain of an existing element;
3. creating or expanding a codelist;
4. adding a new metadata element;
5. adding a new metadata class;
6. imposing a more stringent obligation on an existing metadata element; and
7. imposing a more restrictive domain on an existing metadata element

The profile adds new metadata elements, imposes a more stringent obligation to some existing elements, and imposes a more restrictive domain on some existing metadata elements. Therefore, the following rules for creating the extension are applied:

1. Name, definition, or data type of an existing element shall not be changed.
2. Stringent obligation for existing metadata elements is permitted.
3. Restricting the use of domain values from other metadata elements is permitted.

Following the rules, the option for changing the definition of titles and abstracts is not allowed, and rule three is used instead to restrict the use of domain values from other metadata elements. Table 20 illustrates the profile as an extension of the discovery metadata for geospatial resources of ISO 19115.

As can be seen in the profile, several metadata elements in the discovery metadata were changed from optional (O) to mandatory (M) following the users' requirements (e.g., online links to resources for accessing data samples and data viewer). Some elements from the standard were brought into the discovery metadata such as intended use and previous uses. Two new elements were included for user data ratings and the number of downloads in addition to an element from the standard that could be used for user reviews.

The profile was accompanied by a set of functionalities to present the metadata and develop user interface functionalities. E-commerce or online shopping styles, a criteria- or filter-based search style and a web mapping service for previewing data are the participants' most frequently used interfaces. By implementing these interfaces in the design and development of metadata records and user interfaces, it is expected that the usability of spatial metadata for data discovery and selection will be significantly improved.

**Table 20.** User-oriented spatial metadata profile.

| Metadata Element | Obl. | Occ. | Description |
| --- | --- | --- | --- |
| Metadata reference * | O | 1 | Unique identifier for the metadata |
| Resource identifier * | O | N | Unique identifier for the resource |
| Resource language * | C | 1 | The language and character set used in the resource (if a language is used) |
| Metadata date stamp * | M | N | Reference date(s) for the metadata, especially creation |
| Resource title * | M | 1 | Title by which the resource is known |
| Resource abstract * | M | 1 | A brief description of the content of the resource |
| Resource topic category * | M | N | A selection of the 20 elements in the MD_TopicCategory enumeration that describe the topic of the resource |
| Geographic extent * | M | 1 | Spatial area of the resource |
| Geographic location * | M | 1 | Geographic description or coordinates (latitude/longitude) describing the location of the resource |
| Resource spatial format | M | 1 | A selection of five elements in the MD_SpatialRepresentationType enumeration that describe the method used to represent geographic information in the resource |
| Resource type * | O | 1 | A resource code identifying the type of resource—e.g., data set, a collection, an application (see MD_ScopeCode)—the metadata describes |
| Spatial resolution (equivalentScale/distance) * | M | N | The nominal scale and/or spatial resolution of the resource |
| Positional accuracy | M | N | Accuracy of the position of features |
| Attribute accuracy | M | N | Accuracy of quantitative attributes and correctness of non-quantitative attributes and of the classifications of features and their relationships |
| Level of detail | C | 1 | Brief textual description of the spatial resolution of the resource |
| Data assurance | O | 1 | Information about the outcome of evaluating the obtained value (or set of process stages) against a specified acceptable conformance level |
| Reference date (production) * | M | 1 | A date used to help identify the resource (is existed) |
| Reference date (update) | M | 1 | A date used to help identify the resource (is last updated) |
| Reference date (next update) | M | 1 | A date used to help identify when the resource will be updated |
| Producer | M | N | The organization responsible for creating the resource |
| Intended use | O | 1 | Summary of the intentions with which the resource was developed |
| User ratings | M | N | Users' votes or ratings for the resource based on how it conforms to criteria set by users |
| Number of downloads (use) | M | 1 | The number of occurrences in which the data is used |
| Previous uses | M | N | Brief description of the resource and/or resource series usage (the project) |
| User reviews | M | N | Brief description of the resource and/or resource series usage (reviews against user-specified criteria) |
| Utilization/usefulness | O | N | Textual expression of the descriptive results of conformance to a set of user requirements |
| Thumbnail | M | 1 | Graphical presentation of an area covered by the data or an area where the data is not present |
| Keywords * | M | N | Words or phrases describing the resource to be indexed and searched |
| Resource online link (data sample/viewer) * | M | N | Link (URL) in the metadata for the resource (data sample or Web Map Service) |
| Attribute | M | N | Description of the attribute described by the measurement value |
| Constraints on access and use * | M | N | Restrictions on the access and use of the resource |

**Table 20.** *Cont.*

| Metadata Element | Obl. | Occ. | Description |
|---|---|---|---|
| Use limitation | O | N | Applications determined by the users for which the resource and/or resource series is not suitable |
| Price | O | 1 | Fees and terms for retrieving the resource |
| Lineage * | O | N | A description of the resource(s) and production process(es) used in producing the resource |
| Topology | O | 1 | Degree of complexity of the spatial relationships |
| Resource point of contact * | M | 1 | Name of the person, position or organization responsible for the resource |
| Metadata point of contact * | M | N | The party responsible for the metadata |

* Metadata elements from the original discovery metadata for geospatial resources.

## 5. Discussion and Limitations

This paper used the snowball sampling method to collect data. The aim was to reach out to 10–30 geospatial data users at the global level, following [25]. To this end, the research engaged 61 users from 18 different countries.

The authors acknowledge that controlling the sampling frame in the snowball method is challenging and considered a weakness. We considered the starting sample when contacting our associates to participate in the study and assist us with recruiting more participants. We also used our social media channels and asked our associates to use their channels to recruit participants. We note that the sampling strategy included both social media users and non-social media users. However, our questions to the participants did not include the channel of recruitment. As such, the study lacks insight into how many of the participants are social media users. If most of them are, then the sampling was biased toward a specific group of spatial data users who use social media. However, to the best of our knowledge, there is no study that suggests a correlation exists between being a social media user and having a prejudice toward spatial data portals. As such, even if the bias exists, it does not mean that the results are invalid. We note that the authors are Asia- and Pacific-based researchers. However, the approach was able to engage participants from different continents.

We also note that the number of spatial data users and communities in a given country is not available or, arguably, measurable. The research did not control the number of participants sampled from each county to be proportionate to the number of spatial data users in that country. For the same reason, the research did not control the inclusion of certain countries in the study. We note that we did not target any specific country and the participation in the study was voluntary. As such, we did not have participants from many countries around the world. Participants from more countries could have been included in the study if international bodies such as the OGC were engaged. Having said that, such an approach could have similarly resulted in more knowledgeable and expert participants and fewer novice participants.

The sample provided a depth and breadth of spatial data users for this research. The research engaged 38 expert participants in the work fields of geospatial data production, GIS, or spatial services as well as 21 knowledgeable and two novice users who worked in close professions (e.g., geodesy and land administration) or distant professions (such as psychology and finance). The paper could have achieved a more even distribution of participant types by including professional bodies (e.g., planning, emergency management, engineering) when recruiting the participants.

This paper suggests several additions to the spatial metadata elements for improving the user experience. As this paper argued earlier, spatial metadata systems are often complex from a user perspective. Whether users will find the additional elements discouraging needs to be further investigated.

There will also be implications regarding the introduction of these additional elements and resourcing implications for the organizations that manage spatial data portals. A spatial data portal may have many users on a daily basis. Updating the number of searches for a specific dataset or the number of views of the dataset may demand considerable human and computing resources. Whether the cost of introducing the new elements outweighs the benefits is a commercial and business question for organizations.

## 6. Conclusions

This study identified user requirements and preferred interfaces for spatial metadata based on the results of an internationally circulated survey that targeted spatial data users with differing levels of spatial knowledge and relevant expertise and from a range of professional groups.

The results show that the current and most widely used metadata standard, ISO 19115, provides metadata elements to accommodate most user-required information. However, the standard lacks elements related to user-specified information (i.e., elements for user data ratings and the number of downloads (uses)). An extension is required to accommodate the user requirements for spatial data discovery and selection.

Two new metadata elements were proposed as part of the extension of the standard. The extension was also made to accommodate other required information by changing the type of obligation for some metadata elements. Specific attention was given to the title and abstract elements, where the domains were proposed to change from free text into multiple values from other specified elements. These changes can be inducted into ISO 19115 to transform it into a user-oriented spatial metadata profile. The authors plan to share the proposed amendments with relevant bodies for possible adoption in the following review of the ISO 19115 standard.

**Author Contributions:** Conceptualization, Mohsen Kalantari and Syahrudin Syahrudin; Methodology, Syahrudin Syahrudin; Validation, Syahrudin Syahrudin, Mohsen Kalantari and Abbas Rajabifard; Formal Analysis, Syahrudin Syahrudin; Investigation, Syahrudin Syahrudin; Resources, Mohsen Kalantari; Data Curation, Syahrudin Syahrudin; Writing—Original Draft Preparation, Syahrudin Syahrudin; Writing—Review & Editing, Mohsen Kalantari; Visualization, Mohsen Kalantari; Supervision, Mohsen Kalantari, Abbas Rajabifard; Project Administration, Hannah Hubbard; Funding Acquisition, Mohsen Kalantari, Abbas Rajabifard. All authors have read and agreed to the published version of the manuscript.

**Funding:** This research was funded by the Australian Award Scholarships and the Australian Research Council (grant number DP170100153).

**Institutional Review Board Statement:** The study was conducted according to the guidelines of the Declaration of Helsinki, and approved by the Human Ethics Advisory Group of the University of Melbourne (application number 1545774.3 and 03/10/2018).

**Informed Consent Statement:** Informed consent was obtained from all subjects involved in the study.

**Acknowledgments:** The authors would like to thank all the members of the Center for Spatial Data Infrastructures and Land Administration (CSDILA) and the Center for Disaster Management of the University of Melbourne for all the discussions and enjoyments.

**Conflicts of Interest:** The funders had no role in the design of the study; in the collection, analyses, or interpretation of data; in the writing of the manuscript; or in the decision to publish the results.

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
