# Peer review of "A Proposal for a User-Oriented Spatial Metadata Profile"

_ijgi, doi:10.3390/ijgi10060376_

Round 1
Reviewer 1 Report
The authors have addressed all my previous comments except a few minor comments:
- Some typos in Table 16: “DataIndetification”, “DQ_Descriptivetive”, “spesificUsage”
- Reference 8: DOI could not be found
- Reference 27: URLR does not work - Reference 29: DOI could not be found
Author Response
Thank you for your second review. We appreciate your feedback. Please see attached for our response to your review. The changes are highlighted in the revised version.

Reviewer 2 Report
The manuscript describes a survey study about the usability of metadata and offers insight into how metadata can be improved for usability. The improvements that have been completed by the authors address this reviewer's issues with the earlier draft of the manuscript.
Author Response

(The authors gave the same response as above.)

Reviewer 3 Report
Review of IJGI-885735 v2
The authors strengthened their introductory section to point out that the 2014 ISO standard was updated in 2020 and I appreciate their doing so. I note that in lines 233-234 on page 10, the authors incorrectly state that the latest version of the ISO standard is ISO 19115-1:2014, when in fact it is the amended version ISO 19115-1:2014/Amd 2:2020. The authors state this in their introduction but apparently missed the statement later in the manuscript. This is easy to fix, of course. Many other comments in the initial review appear to have been largely ignored.
Some grammar and syntax issues persist, for example line 42 on page 1 should read “…create and update metadata records” (plural, not singular). I see that some errors have been corrected however there are still issues throughout the paper, and I encourage the authors to have a native English speaker read the next draft to ensure legibility.
The snowball sampling strategy has not been revised, although the discussion of bias requested in my initial review is partially addressed in an expanded section 5 of the manuscript. Lines 350-352 on page 16 make two statements that are incorrect. First, the authors state that they considered the “distinctive characteristic” of the starting snowball sample. This would have been impossible given that the survey was distributed across social media and then snowballed, since one cannot necessarily control who looks at the social media page. Second, the use of social media in and of itself does not constitute random sampling of all spatial data users, given that many individuals (in and beyond the spatial data community) refuse to use social media of any type. Thus the sampling strategy is biased from the outset to those who still use social media, and this is a major limitation of the study. At a minimum, respondents should have been questioned to determine who accessed the survey independently and who received it from a colleague. This would allow researchers to flag respondent answers for potential bias. The authors need to discuss all of these problematic aspects of their research design, but they do not do so.
In my first review, I also asked for expansion of the discussion in Table 1: “In Table 1, I was unclear on item 3, the searching criteria and presentation style. What specific questions did the questionnaire cover here? I use very different search criteria for each spatial data search I undertake and would be unable to give a short answer to the question “what search criteria do you use?”. “ This request has not been addressed. The second part of my request referred to the meaning of presentation style. While the authors substituted “User Interface” that term in row 3 of the table, they do not explain that term (nor presentation style) in the text. So now I have three questions about Table 1 that need clarification in the text: explain what is meant by searching criteria, user interface. I note also that the explanation of the term “presentation style” could have been moved forward in the text from Table 8, but this has not been done in the revision.
Figure 1 does explain the semantics of the graduated symbols in its caption but there is still nothing in the text. Symbol placements remain awkward. For example, why is the Africa icon at the southern tip of Africa, and why is Mozambique also shown? Africa is not a single country so perhaps this respondent comes from South Africa, which is a country. The placements of Canada and USA icons raise similar questions. The blurry version of Figure 2 has been removed from the manuscript, with written text introduced, and this has improved the discussion.
Another concern relates to my initial request to explain why the sampling strategy claims many respondents from outside the spatial data industry. Lines 99-103 on page 4 indicate that this concern has also not been addressed by the authors. For example, professional societies could have assisted in disseminating the survey (I asked about this). While I do not expect the authors to re-run the survey, it is surprising that the text does not include even a short statement about why this option was not utilized.
The discussion section has been added to expand on some limitations of the research methodology but the statements are either incorrect (as I state above) or articulated in such a weak fashion as to make it appear that the authors don’t actually take seriously the problems evident in their methodology. Although they do admit that the sample size and distribution is not proportional to the number of users in each county, they neglect to address the absence of any response from countries with known high spatial data usage (e.g, Russia, much of western Europe, Mexico, Japan, China, United Kingdom, Scandinavia). This result by itself should have alerted the authors that their decision to distribute the survey only through social media was by itself a limiting factor in the analysis.
Author Response

(The authors gave the same response as above.)

Round 2
Reviewer 3 Report
I believe the authors have addressed my concerns. The English language and syntax are much improved as well. In spite of continuing concerns about the bias introduced by the sampling strategy, it is clear that the authors have made a sincere attempt to address limitations in the manuscript.
This manuscript is a resubmission of an earlier submission. The following is a list of the peer review reports and author responses from that submission.
Round 1
Reviewer 1 Report
This paper is a valuable and original contribution to make geospatial metadata more user driven. To that end the authors have conducted a survey among 61 users of spatial data portals in 18 countries and collected requirements. Based on the results of the survey they propose an extension of the ISO 19115-1:2014 metadata standard and some adjustments, e.g. in the optional/mandatory nature of some of the elements. The objectives of the study are well introduced, the paper is well structured, and the results of the survey are presented in detail through a series of tables and discussed. Literature review is well conducted. The paper could however benefit from improvements that are suggested below.
Major comments
- It would be interesting to know which geospatial data portals have been used by the participants of the survey. Since this paper is about improving access to data from metadata but also from a user-experience perspective, this information could be useful to better understand on which basis users’ needs were collected.
- In my view it is important that the authors explain how they intend to present and discuss the proposed metadata profile extension to/with the persons in charge of implementing metadata standards
- It is well-known that metadata production is still perceived by users as a complex, tedious and time-consuming task. This typically results in little metadata production and seriously hinders the objective of facilitating data discovery. In the manuscript, the authors propose to add two new metadata elements and to change the status of some other elements from optional to mandatory. A discussion is missing here about the fact that this might discourage even more users from filling up metadata
- Can the authors provide the list of questions in addition to the structure of the questionnaire?
Minor comments
- Line 14: “discovery and use of spatial data” => “discoverying and using spatial data”
- L15: “SDI” => “spatial data infrastructure” (do not use acronym)
- L16: What does “spatial metadata data” mean?
- L28: missing word “been” in “have predominantly designed”
- L30: what is the difference between “spatial data” and “datasets”?
- L33: please remove “numerous”
- L38: “with specific knowledge”: it is not clear to me what the authors mean
- L50: it is not clear at this stage which surveys the authors are referring to. If this is the survey that conducted for this research, can the authors please introduce it here and not at the beginning of the “Research method” paragraph?
- L53-54: I suggest replacing “of the discovery metadata of the standard” with “of the standard profile”
- L65: can the authors specify how many participants were validated and how many were rejected?
- L71: “discovery” => “identification”
- Table 2: I am wondering whether some participants had the possibility to select two or more options. For example, a participant can work in disaster management, but also in GIS. If this was not the case, I think this would deserve to be mentioned
- L104: “As, these users will potentially grow in number over time”: how do the authors know this? Can they please include a scientific reference or remove this sentence?
- L108: “As well as” => “In addition to”
- Table 8: what does “occasional” and “regular” mean? Can the authors be more explicit?
- Table 8: “free hand boundary”: can the authors be consistent with the terminology used previously (“freehand drawing tools”)?
- Table 8: “Filter based presentation as in e-commerce” is duplicated
- L164: “and selected” => “and how data can be selected”
- L200: “As well as” => “In addition to”
- L235: “web map service browser” => “web map service capability”
- Figure 2 is difficult to read. Can the authors make it more readable?
- Caption of Figure 2 should read “Metadata schema classes” instead of “Metadata schema”
- Some typos in Table 16: “DataIndetification”, “DQ_Descriptivetive”, “spesificUsage”
- L265-266: language is not good. Can the authors rephrase?
References
- Reference 1 is not complete
- DOI in reference 6 (Boin and Hunter) could not be found
- Reference 8: can the authors remove the first name of Devillers; DOI could not be found
- Can the authors cite Goodchild as follows (reference 13): Goodchild, M. F. (2008, June). Spatial accuracy 2.0. In Proceedings of the eighth international symposium on spatial accuracy assessment in natural resources and environmental sciences (Vol. 1, pp. 1-7).
- Same for reference 14: Goodchild, M. F. (2007, June). Beyond metadata: Towards user-centric description of data quality. In Proceedings, Spatial Data Quality 2007 International Symposium on Spatial Data Quality, June (pp. 13-15).
- URL in reference 26 (Olfat) does not work
- Reference 27 (Olfat et al.): DOI could not be found
- Reference 29: can the URL be replaced by: In GSDI 12 World Conference: Realising Spatially Enabled Societies, Singapore
Reviewer 2 Report
The described usability study has the potential to inform the Earth science community about preferences for producing and representing metadata records that describe Earth science data products. The study also offers insight into preferences for user interfaces that would provide access to such data as well as providing recommendations for other information that might be offered to users of Earth science data to improve discovery and use.
The manuscript also offer recommendations for improving current metadata standards and for including new elements and rules for such standards.
The manuscript contains very many typographical and semantical errors. These errors appear throughout the manuscript, indicating the need for comprehensive review of the manuscript by the authors. There are too many errors to be described here and such error do present a distraction for readers and must be corrected. Some errors that appear at the beginning of the manuscript, along with some suggested corrections, are listed as examples, below:
Line 28: Replace "have predominantly designed" with "have been predominantly designed".
Line 29: Remove "of".
Line 39: Finish sentence so that it answers the question, 'puts further pressure on what?'.
Line 40: Add "a" before "strong".
Line 42: Change "record" to "records".
Relatedly, the Author Contributions section should not include the instructions for authors to complete that section.
Since the study represents research on human subjects and describes the methods and results of the survey, the manuscript should describe the approval that the survey study received from an Institutional Review Board (IRB) or ethics board for the protection of human research subjects, stating the particular IRB that has approved the research. This information should be placed in the Acknowledgements section or described in the Methods section.
Critically, the manuscript appears to be missing a section on Limitations, which is customary for manuscripts that describe the methods and results of a survey study.
Among other limitations, the manuscript should describe how snowball recruitment of survey respondent can introduce sampling bias and cite the relevant literature. Limitations in terms of small sample size and, in particular the limited number of respondents designated as experts, also should be described, as well as other limitations, based on the literature on conducting surveys.
Reviewer 3 Report
I found the analysis done based on an improvement in the usability of metadata interesting. I also found interesting the number of participants, their different expertise and the geographical area of origin
Section 3.2 does not clearly understand what is sought. The argument is based on an article [33], which is not accessible (“Access Status: This item is embargoed and will be available on 2022-03-18”) and therefore I have not been able to verify it. I think it is necessary to explain better the meaning of the headings of Tables 5, 6 and 7, in particular the meaning of “Interface style”
Line 206 says: "The results show that only participants from the expert group chose to give responses to the open-ended questions." I think it should say: "It should be noted that only the experts gave their opinions on the open-ended questions" (or am I wrong?
Since usability is talked about, and this concept is closely related to visualization, graphic examples (images) should be used to facilitate some of the statements made in the text.Line 264. The legend of Table 16 is not complete. In particular, the heading of the third column where different concepts appear should be better explained: M / 1, C / N, O / N, C71, M / N, which are not sufficiently explained at the foot of the page
Table 16: It is not easy to visualize the correspondence between the elements of the three columns, so it is recommended to incorporate the horizontal lines that separate the rows of the table
And finally, something very debatable outside the evaluation of the article. I think that soon there will be more and more users of the geodata information metadata. And many of them will be totally foreign to the academic world, so if UML-type schemes are multiplied in the explanations, it will lead to users fleeing that document because that is not understood by ordinary people. Wouldn't there be kinder (and more aesthetic!) Metadata information than using UML schemas? Perhaps an article like the current one, which deals with the usability of metadata, should slightly point this question ... (sorry for this intrusion but your article talks about non-expert users ...)
Reviewer 4 Report
This paper argues for two new metadata elements to be added to ISO 19115-1:2014. Many users do not fully understand metadata standards, their content or necessity. And on this point, the paper is timely and relevant. The discussion in the manuscript weakens the authors’ stated objective however. A rewrite should strengthen their arguments considerably.
All ISO standards are reviewed every 5 years, and this one was reviewed and re-accepted in 2019 and thus remains current. This point could be brought into the Introduction to help the multitude of spatial data users understand that the standard remains current in spite of the 2014 label implying a standard that is 6 years old.
I commend the authors’ decision to survey spatial data users globally. I would like to see further discussion about how a snowball strategy “maintains the validity” of participants. Snowball sampling is a workable strategy when a subject pool is not immediately available, or unknown to the researchers. The downside of snowball sampling is the loss of random independent sampling, meaning that there is no way to protect against possible bias. In passing the survey to one’s colleagues, the sample might become biased toward a particular viewpoint about the subject. So how did these researchers ensure that all perspectives were sampled, that people were sampled who are satisfied with the standard, or see no need for extensions? How is the sampling frame controlled to ensure that the number of participants sampled from each part of the world is commensurate with the number of spatial data users in each part of the world? That is the only way to claim that the sample represents a global viewpoint, as the authors claim. At a minimum, these sampling frame issues need to be discussed, instead of asking readers to accept on blind faith that the snowball sample represents the full continuum perspectives being studied.
In Table 1, I was unclear on item 3, the searching criteria and presentation style. What specific questions did the questionnaire cover here? I use very different search criteria for each spatial data search I undertake and would be unable to give a short answer to the question “what search criteria do you use?”. Also, what is meant by “presentation style”, is this the way a data portal presents the data, or is it the way that the participant presents a query, or is it their preference for a data portal interface? Table 5 does not clarify this question, listing a particular Interface, but not explaining which of the three options one could interpret? Adding some explanatory text to the discussion would easily clarify.
In Figure 1, what is the meaning of the different sized icons, are these proportional circles? The figure lacks a legend that could clarify. If the symbols are proportional, they certainly do not reflect the global demographics of spatial data user communities. For example no participants from the UK or France are shown, nor from Scandinavia, PRC China or Taiwan, where one might expect very large user communities. Hence my concern about possible bias.
The authors claim that the respondents come from diverse backgrounds (and the data seems to agree). They also claim however that the survey pool contained “many spatial data users outside the spatial industry” and yet in my reading of the 19 fields of expertise, I saw no categories listed that lie outside. Table 3 confirms this point, listing only 2 of 61 participants who claim no experience with spatial data. Herein lies a second limitation of snowball sampling, that large sectors of the target audience might be missed simply due to the cascading of sampling requests only through known colleagues. A better choice might be to work with international professional societies such as AGILE or CaGIS or INSPIRE or ANZLIC (which the authors mention but did not utilize to help set up the survey).
In the manuscript on page 6 is listed a set of five elements needed by users searching a data portal. The ISO standard includes all of these elements. Thus any data portal adhering to the standard would include these elements in its metadata. At this point in reading the manuscript, I lost sight of the objective of the survey. Reading on, I arrived at Table 8, which finally clarified the “presentation style” question I raised earlier. By the way, Table 8 contains a duplicate entry for “bounding box for location search”.
Figure 2 showing the metadata schema is quite blurry and hard to read. I encourage the authors to submit images at a finer resolution, and to acknowledge the source, weblink and page number of images taken directly from the ISO standard.
The proposal to add specific information on number of downloads might be realistic for a data portal to monitor and report, although such reporting might not be as timely as some users might want. A National Mapping Agency portal for example gets thousands of searches daily or even hourly, and updating this even weekly or monthly and for all data files on the portal might become an expensive task, even when automated, as it would take computer processing time away from other portal tasks. Collecting user reviews or a user data rating would require even more processing and portal resources (e.g., tagging user reviews to a particular data set and version). For larger data producers particularly, the authors’ proposals seem unfeasible. For smaller data producers, the proposals seem insensitive to the amount of effort and resources needed to create and steward a data portal. I do want to acknowledge that the authors took the time to go through the existing ISO standard to determine how to propose an extension.
As a final comment, the text is quite challenging to understand. Some English syntax issues are readily corrected, e.g., in the abstract, “bringing them into the discovery” should be “bring them…”. In the first sentence of the Introduction, “… standards have BEEN predominantly …” (missing verb participle). Other sentences are quite confusing, as for example (again in the Introduction): “… the needs and requirements of non-expert spatial data puts further pressure.” is unclear. First off, on whom is the further pressure being put? Secondly, the first half of the sentence refers to users with “specific knowledge” which I assume means they are somehow experts. I don’t understand why being in or outside the field makes a difference in the sentence as the domain of expertise is undefined. Third, I do not understand the concept of “non-expert spatial data” since data by itself carries no particular expertise – do the authors mean “non-expert data users”? And if so, how do these users relate to those with “specific knowledge”? I’m not trying to be obstructionist here. My point is that sentences like this one make it very difficult to understand the points the authors are trying to make, and there are several of these throughout the manuscript, which makes it hard to capture the justification for the proposal. I cannot tell if the problem is a lack of understanding or a problem with language. I assume the latter is true, and urge the authors a have a native English speaker read the text to clarify and correct problems.
